# NUT Is a Driver of p300-Mediated Histone Hyperacetylation: From Spermatogenesis to Cancer

**DOI:** 10.3390/cancers14092234

**Published:** 2022-04-29

**Authors:** Sophie Rousseaux, Nicolas Reynoird, Saadi Khochbin

**Affiliations:** Institute for Advanced Biosciences, Université Grenoble-Alpes, INSERM U1209, CNRS UMR 5309, 38706 La Tronche, France; sophie.rousseaux@univ-grenoble-alpes.fr (S.R.); nicolas.reynoird@univ-grenoble-alpes.fr (N.R.)

**Keywords:** NUT midline carcinoma, NUT carcinoma, spermatogenesis, BRDT, BRD4, BRD3, bromodomain, p300, CBP, NUTM1, histone hyperacetylation, protamine, histone-to-protamine replacement, acylation, butyrylation, crotonylation

## Abstract

**Simple Summary:**

The functional characterization of the BRD4-NUT fusion protein as the driver of the highly aggressive NUT Carcinoma is fundamental to the understanding of the mechanisms responsible for the genome-wide hyperacetylation of histones prior to their eviction during the final stages of sperm cells maturation.

**Abstract:**

In maturing sperm cells, a major genome re-organization takes place, which includes a global increase in the acetylation of histones prior to their replacement by protamines, the latter being responsible for the tight packaging of the male genome. Understanding the function of the oncogenic BRD4-NUT fusion protein in NUT carcinoma (NC) cells has proven to be essential in uncovering the mechanisms underlying histone hyperacetylation in spermatogenic cells. Indeed, these studies have revealed the mechanism by which a cooperation between BRD4, a bromodomain factor of the BET family, NUT, a normally testis-specific factor, and the histone acetyltransferase p300, induces the generation of hyperacetylated chromatin domains which are present in NC cells. The generation of *Nut* ko mice enabled us to demonstrate a genetic interaction between *Nut* and *Brdt*, encoding BRDT, a testis-specific BRD4-like factor. Indeed, in spermatogenic cells, NUT and p300 interact, which results in an increased acetylation of histone H4 at both positions K5 and K8. These two positions, when both acetylated, are specifically recognized by the first bromodomain of BRDT, which then mediates the removal of histone and their replacement by protamines. Taken together, these investigations show that the fusion of NUT to BRD4 in NUT Carcinoma cells reconstitutes, in somatic cells, a functional loop, which normally drives histone hyperacetylation and chromatin binding by a BET factor in spermatogenic cells.

## 1. Introduction

NUT (Nuclear protein in Testis) is a testis-specific factor originally discovered as a chromosomal fusion partner of BRD4 and BRD3, both members of the BET double bromodomain-containing family of proteins, in an aggressive cancer known as NUT Carcinoma (NC) [1,2].

These observations suggested that NUT could be a central element in the underlying oncogenic molecular mechanisms. However, besides the specific expression of its encoding gene in testis, nothing was known on the physiological and pathological functions of NUT. The presence of members of the BET (Bromodomain and Extra-Termina) family, especially BRD4 (bromodomain containing 4), in fusion with NUT in a large number of NC cancer cases, suggested that the oncogenic activity driven by the BRD4/3-NUT fusion proteins could involve a yet unknown cooperation between the two fusion partners. More specifically, since the proteins of the BET family are known acetylation readers, this strong bias towards bromodomain factors among all the identified NUT fusion partners, suggested that histone acetylation and acetylation reading factors could be important components of the underlying oncogenic mechanism [3]. Furthermore, following this work, and because of its physiological context of expression in male germ cells, the hypothesis was made that NUT could also play a role in histone hyperacetylation associated with genome-wide histone-to-protamine replacement, which occurs in the late phases of spermatogenesis [4].

Histone-to-protamine replacement is one of the most spectacular known large-scale genome packaging reorganizations known in eukaryotes [5]. Despite its dramatic nature and its essential role in procreation and species perpetuation, the molecular basis of this remarkable and unique genome reorganization has remained a black box in biology for many years.

Early investigations, which aimed at deciphering the molecular basis of these events using mouse models, revealed that the process of histone-to-protamine replacement is associated with the occurrence of a wave of histone hyperacetylation in mice [6], as well as in various other species (please see [7], and references therein).

After the ground-breaking discovery that bromodomains are binders for acetylated histones by Ming-Ming Zhou and colleagues [8], our laboratory hypothesized that bromodomain-containing factors could also mediate events which are dependent on histone hyperacetylation in post-meiotic haploid spermatogenic cells (or spermatids) in relation to histone-to-Protamine replacement.

Following this hypothesis, an in silico approach was used to identify putative testis-specific bromodomain-containing factors as possible candidates capable of acting on hyperacetylated histones in spermatids. This strategy identified *Brdt* (bromodomain testis-specific), a gene encoding a testis-specific member of the BET family, whose function was completely unknown at that time, as a possible candidate [9]. Functional investigations that followed revealed the capacity of ectopically expressed BRDT to dramatically compact and reorganize chromatin in response to chromatin hyperacetylation induced by histone deacetylase (HDAC) inhibitor, TSA, treatment [9,10]. Further structural studies showed that BRDT’s first bromodomain (BD1) presents the remarkable property to specifically bind histone H4 tail bearing simultaneous acetylation at K5 and K8 [11]. Since the co-acetylation of these two lysines is a signature of hyperacetylated H4 [12,13], these studies reenforced our hypothesis that BRDT could be a factor acting on hyperacetylated chromatin at the time of histone-to-protamine replacement. This ability of BRDT to bind hyperacetylated H4 was also confirmed in vivo [14] and later BD1 was also shown to bind nucleosomal DNA, in addition of H4K5acK8ac, increasing the stability of its interaction with hyperacetylated H4-bearing nucleosomes [15]. Finally, by investigating various mouse *Brdt* genetic models, our laboratory confirmed the role of BRDT’s BD1 in the removal and replacement of histones in spermatids [10].

However, despite this progress in our understanding of the connection between H4 hyperacetylation and histone-to-protamine replacement, the origin of the histone H4 hyperacetylation wave has remained obscure.

In order to obtain a full understanding of the process of this acetylation-dependent histone eviction, it appeared of crucial importance to discover the origin of spermatid-specific H4 hyperacetylation.

Following a series of unsuccessful attempts in our laboratory to find mechanisms underlying H4 hyperacetylation in spermatids, we thought that functional studies of the fusion protein BRD4-NUT in cancer cells could shed some light on the connection between histone acetylation and acetylation-dependent events in spermatids. Indeed, since NUT, a testis-specific factor of unknown function, is fused to and cooperates with BRD4 in the context of NC, the hypothesis was made that, in its physiological context, NUT cooperates with BRDT, and that the NC chromosomal translocation observed in the context of somatic cells cancer actually re-establishes this cooperation.

Hence, functional studies of BRD4-NUT were undertaken [3] with the hope that the understanding its function in the context of NC would also provide clues to fully understand the role of histone hyperacetylation in spermatids.

This reasoning turned out to be correct, since these early investigations of the BRD4-NUT fusion protein allowed our laboratory to unravel the mechanism by which this functional cooperation between NUT and BRD4 could support the oncogenic activity of the fusion protein. Indeed, the data obtained showed that BRD4-NUT drives chromatin hyperacetylation through a feedforward loop, resulting in the generation of hyperacetylated chromatin foci. This loop involves an increased histone acetylation by an interaction between NUT and the histone acetyltransferase (HAT) p300 and the binding of this acetylated chromatin by BRD4 [3] (Figure 1).

These data also strongly suggest that NUT could be an excellent candidate factor, which could drive the observed histone hyperacetylation associated with histone replacement in spermatids.

A research program was subsequently developed to investigate the role of NUT in the histone hyperacetylation wave observed prior to histone removal in the physiological context of haploid male germ cells.

## 2. NUT Is Specifically Expressed in Post-Meiotic Phases of Spermatogenesis

Spermatogenesis is a highly specialized differentiation program encompassing three characteristic stages, which include mitotic (or pre-meiotic), meiotic and post-meiotic phases. In the testis, a population of diploid adult stem cells, known as spermatogonia, is either maintained as stem cells or committed to differentiation following mitotic divisions. The committed spermatogonia-derived cells become spermatocytes while they undergo two meiotic divisions to generate haploid post-meiotic cells named spermatids.

The post-meiotic maturation of spermatids involves several major morphological and functional changes including a genome-wide chromatin remodelling and genome reorganization, resulting in the extreme compaction of the male genome in mature sperm cells [5,16,17]. Indeed, in these cells, the universal mode of eukaryotic chromatin organization, based on units named nucleosomes (each nucleosome consists of an octamer of 4 core histones, H2A, H2B, H3 and H4, around which the DNA is wrapped) shifts toward a new genome packaging structure based on the association of DNA with non-histone small basic proteins called protamines [5]. This histone-to-protamine transition allows a very tight compaction of the genome in mature spermatozoa, which is essential to protect the paternal genome during its transportation out of the parent organism through harsh environmental conditions in order to reach the female gamete, the oocyte [16].

The molecular basis of the essential process of histone-to-protamine replacement has remained one of the most obscure of all biological phenomena. The initial knowledge of the molecular basis of histone-to-protamine replacement was limited to only a few facts including a genome-wide hyperacetylation of histones [7] prior to their replacement and the expression of several highly specific histone variants by spermatogenic cells [17]. However, when this project was started in our laboratory, nothing was known about the mechanisms driving this histone hyperacetylation, or its role in histone eviction, or the role of histone variants.

The early published work on the activity of the BRD4-NUT fusion protein [3] identified NUT as an excellent candidate in driving the histone hyperacetylation associated with histone eviction in spermatids, which prompted our laboratory to specifically consider its role in this process during mouse spermatogenesis.

This hypothesis was reenforced by considering the pattern of NUT expression during spermatogenesis. Indeed, at the mRNA as well as protein levels, NUT first appears in post-meiotic cells, just at the time when histone hyperacetylation starts [4].

## 3. NUT Is Essential for Histone H4 Hyperacetylation and Histone-to-Protamine Replacement

To test our hypothesis, *Nut* ko mice were generated. *Nut* ko male mice turned out to be infertile, with a total absence of spermatozoa. A more detailed analysis of spermatogenesis demonstrated that spermatids disappear at the time of protamine assembly and histone displacement [4]. This observation suggests that the absence of NUT could create an acute cell toxicity when cells prepare to set up the process of histone-to-protamine replacement.

Because of a possible role for NUT in histone hyperacetylation, an unbiased quantitative and qualitative proteomic analysis comparing the acetylation levels for each histone lysine position between wild-type spermatid cells expressing NUT and their *Nut* ko counterparts was carried out. Remarkably, this proteomic analysis, which was confirmed by immunoblotting, demonstrated that NUT is required for the acetylation of histone H4, specifically at K5 and K8. Since, as previously mentioned, the co-occurrence of H4K5acK8ac is known as a signature of hyperacetylated H4 and that it is required for the binding of BRDT itself involved in downstream events [10], these results supported the hypothesis that NUT is a critical actor in mediating the observed histone hyperacetylation in spermatids at the time of histone eviction [4].

## 4. Genetic Interaction between NUT and BRDT

An unbiased histone acetylome analysis identified H4 lysine 5 and lysine 8 (H4K5K8) as particularly sensitive to NUT-mediated H4 acetylation [4]. Considering that the previous structural studies of BRDT’s bromodomains demonstrated that BRDT’s BD1 precisely recognizes H4K5acK8ac [11], these new results appeared very relevant and exciting.

This observation suggested that BRDT’s action in spermatids would be dependent on the prior action of NUT, and therefore the prediction was that spermatids from *Nut* ko mice should show a phenotype similar as that of spermatids from *Brdt* delta-BD1 mice, expressing a BRDT mutant protein deleted for its first bromodomain (BRDT delta-BD1). Previous investigations had shown that spermatids expressing BRDT delta-BD1 are unable to replace their histones [10]. In these cells, although protamines are normally expressed, they are not incorporated and instead they accumulate around the nucleus [10,18]. Remarkably, exactly the same phenotype is observed in *Nut* ko spermatids. Indeed, these cells express protamines but, as observed in *Brdt* delta-BD1 spermatids, these protamines remain around the nucleus and the histones are not displaced [4].

Hence, these observations strongly support the hypothesis that NUT and BRDT function along the same molecular pathway, leading to histone-to-protamine replacement. The requirement of NUT to induce the acetylation of H4 on K5 and K8 and the ability of BRDT BD1 to specifically recognize and bind H4K5acK8ac and to displace histones, perfectly explain the observed genetic interaction between NUT and BRDT (Figure 2).

## 5. NUT-Mediated Histone H4 Hyperacetylation by p300

In vitro experiments using purified p300, histone octamer as a substrate, and the subsequent histone acetylome analysis, showed that, as expected, p300 preferentially acetylates H3, particularly at positions K18, K23 and K27, with limited activity on H4 tail lysines [4]. Upon the addition of a purified NUT fragment capable of interacting with p300, H4 K5 and K8 also become remarkably acetylated [4]. This in vitro experiment nicely reproduces the in vivo situation, where the expression of NUT is associated with a clear enhancement of H4K5K8 acetylation. Additionally, the immunoprecipitation of NUT and the proteomic analysis of its associated proteome, confirmed the association of both p300 and CBP with NUT in spermatids. Therefore, from these experiments, it is possible to confirm that the activation of the *Nut* gene expression in spermatids, and the recruitment of CBP/p300 by NUT are major driver elements in inducing histone H4 hyperacetylation and histone replacement.

The functional interaction between NUT and CBP/p300 was further supported when the transcriptomic of *Nut* ko cells was compared to the transcriptomic of a mouse model showing a slight decrease in CBP/p300 expression in their spermatids. Indeed, a significant overlap was observed between genes affected by CBP/p300 down-regulation and genes affected by the absence of NUT [4]. CBP/p300 protein down-regulation in spermatids was obtained following a conditional ko of *CBP* and *p300* in post-meiotic spermatogenic cells [19].

Based on these studies it is possible to propose that histone hyperacetylation in spermatids is directed by the ubiquitously expressed HATs, CBP and p300, which are present at all stages of spermatogenesis [19]. The specific activation of NUT in spermatids allows a switch of CBP and p300 towards a new activity, which is the targeted acetylation of H4 k5 and K8, creating binding sites for BRDT at a genome-wide scale (Figure 2).

## 6. Conclusions and Future Perspective

The discovery of BRD4-NUT as a p300 mobilizing hyperacetylator machinery [3] shed light on the still obscure mechanism of histone hyperacetylation during late spermatogenesis [4].

In both NUT carcinoma cells and post-meiotic male germ cells, these investigations highlight a role for NUT in directing a p300-dependent histone acetylation and chromatin-binding by a bromodomain containing factor.

In spermatids, it is the NUT-p300 complex that directs a genome-wide histone H4 hyperacetylation, providing binding sites for BRDT’s first bromodomain BD1, which acts downstream of this acetylation wave. Interestingly, the three actors are also at play in NUT carcinoma cells, involving the expression of the BRD4-NUT fusion protein in the generation of foci of hyperacetylated chromatin characterizing NC cells. In this case, after the initial binding of BRD4-NUT to an acetylated chromatin site (nucleation phasis), NUT recruits p300, leading to local hyperacetylation and the recruitment of additional BRD4-NUT, initiating a feedforward mechanism, which results in the spreading of chromatin acetylation and BRD4-NUT binding (Figure 1). In the case of BRD4-NUT, the fusion of the two proteins creates an obligatory cooperation between BRD4 and the NUT-p300 complex. In spermatids, a similar cooperation between NUT-p300 and BRDT is observed, but a major difference is that the three actors are free and are not bound to stay together (Figure 2).

The question is why, in NC, the fusion of BRD4 and NUT appears as an obligatory event in ensuring its oncogenic activities. One reason is that NUT is a testis-specific gene and, hence, the chromosomal fusion allows for its expression under the ubiquitous BRD4 promoter. The second reason could be that, if BRD4 and NUT were not fused, although NUT could recruit and activate CBP/p300, there would be no anchoring point for this complex, since NUT does not interact with BRD4; the free NUT-p300 complex would be diluted all along numerous CBP/p300 histone and non-histone substrates.

The oncogenic role of an anchoring point for the localized recruitment of the NUT-p300 complex could also be at play in other NUT fusion oncogenes, involving a variety of non-BET partners [20]. In most of these cases, the NUT fusion partners are chromatin-binding and transcriptional regulators and, hence, even if they cannot ensure the hyperacetylator function of the BRD4-NUT fusion protein, they would be able to sequester CBP/p300 at the anchoring points, leading to a global histone hypoacetylation outside of the recruitment points. Indeed, in the case of BRD4-NUT, the sequestration of p300 in BRD4-NUT foci was previously shown to impair p53 acetylation signalling [3] and to lead to a general histone hypoacetylation due to the sequestration of CBP/p300 in the BRD4-NUT foci [21].

Therefore, the sequestration of CBP/p300 by the NUT fusion oncogenes could lead to the impairment of the CBP/p300 signalling process, which could contribute to the underlying oncogenesis.

In the case of the BRD4-NUT fusion, an interesting general conclusion of these investigations is that a specific mechanism, involving a cooperation of two factors, a BET factor and NUT, that normally operates in the very particular context of spermatids, has been awakened in somatic cells due to a chromosomal translocation event.

Our full understanding of the oncogenic activity of BRD4-NUT and the underlying mechanism now allows us to envision specific therapeutic strategies based on the inhibition of BET bromodomains, CBP/p300 and HDACs by cell permeable small molecule inhibitors.

The work on spermatogenesis and functional investigation of BRDT has also revealed another property of BET factors that could be exploited to improve the response of the BRD4-NUT fusion protein to specific treatments, specifically to BET inhibitors.

Indeed, it is now clearly established that, in addition of acetylation, histones could be modified by a series of competing acylations including propionylation, butyrylation, crotonylation, succinylation, etc. [22]. Investigations from our laboratory and other groups showed that most of the bromodomain-containing factors, and BETs in particular, are unable to bind histones modified with longer chain acyl groups [23,24,25,26]. Accordingly, we observed that H4-bearing butyrylation at K5 (H4K5bu) escapes the BRDT-dependent histone removal in spermatids [23].

Based on these data, it is possible to hypothesize that a dynamic exchange between acetyl and acyl groups at position H4K5 should destabilize the binding of BRDT or other BETs to chromatin and increase their dynamics. A stable and permanent acetylation would immobilize BET factors on chromatin, while a permanent longer chain acylation (butyrylation, crotonylation) would inhibit BET factor binding. Both of these situations would be associated with decreased activity of these factors.

Finally, using the context of B Acute Lymphoblastic Leukaemia (B-ALL), it has recently been demonstrated that the metabolism-driven H4K5 acetylation/acylation ratio is able to tune the stability of the binding of BRD4 with chromatin [26]. These data showed that a metabolic activity that favours histone H4K5 butyrylation and crotonylation over acetylation increases BRD4 dynamics and sensitivity to the BET inhibitor JQ1. By using a BRD4-NUT expressing cell model, it was indeed possible to directly show that an increase in the butyrylation-crotonylation/acetylation ratio increases the motility and dynamics of BRD4-NUT chromatin binding.

Based on these observations, we can predict that cell metabolic pathways that favour the long chain acylations of histones would increase the response of BRD4-NUT to BET inhibitors (Figure 3).

Overall, these investigations highlight how the deep understanding of the biology of BET factors, NUT and CBP/p300 and the consideration of their physiological context of action, could help understanding their involvement in pathologies, specifically cancer, and vice versa.

In our case, it was the willing to decipher the molecular basis of histone H4 hyperacetylation in the physiological context of the histone-to-protamine replacement in male germ cells that led us to discover the functional interconnexion between BRD4, NUT and p300 in the context of NUT Carcinoma cells [3].

In many other cases of somatic cancers where testis-specific genes are aberrantly expressed [27], the knowledge of their physiological role in male germ cells could similarly shed light on new oncogenic mechanisms and give a basis for the development of specific therapeutic strategies.

## Figures and Tables

**Figure 1 cancers-14-02234-f001:**
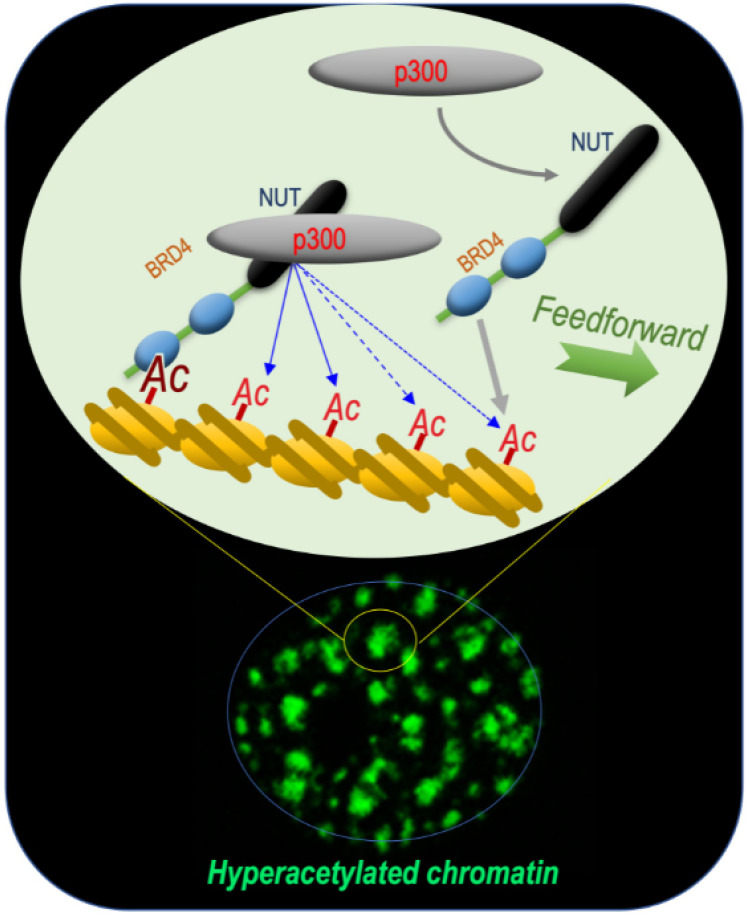
BRD4-NUT and p300 cooperate to induce hyperacetylated chromatin domains through a chromatin acetylation/binding loop feedforward mechanism. The mechanism schematized is based on data published by Reynoird and colleagues [3]. The scheme represents our understanding of the mechanism underlying the initiation and propagation of hyperacetylated chromatin domains leading to the creation of multiple BRD4-NUT/p300-hyperacetylated chromatin foci. The initiation of the process corresponds to the binding of acetylated nucleosomes by BRD4, the recruitment of p300 by NUT and the enhancement of p300 catalytic activity, leading to the acetylation of adjacent nucleosomes. These new sites of histone acetylation recruit additional BRD4-NUT molecules, which in turn recruit p300 and a feedforward loop of histone acetylation-BRD4-NUT binding starts. The BRD4-NUT/p300 dependent histone acetylation propagation encounters opposing deacetylase activities constraining the acetylation propagation. The limits of the acetylated chromatin foci are dynamic and could move forward or backward depending on the opposing activities of acetylation and deacetylation.

**Figure 2 cancers-14-02234-f002:**
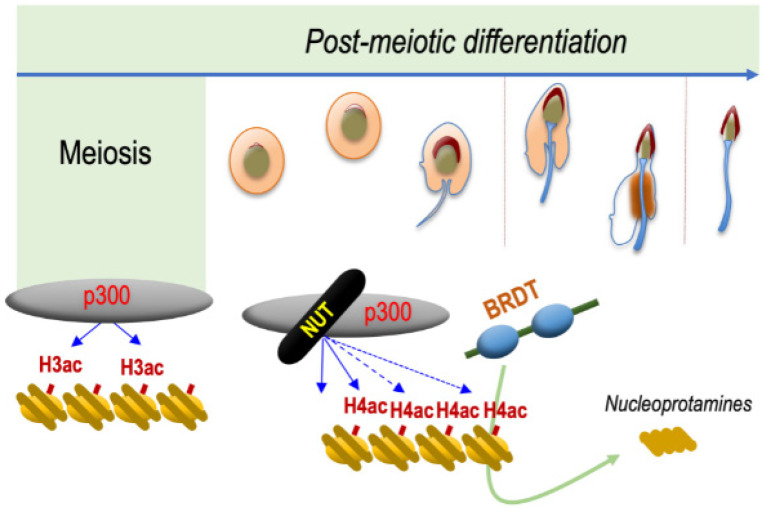
NUT is specifically expressed in post-meiotic spermatogenic cells and drives histone H4 hyperacetylation preceding histone eviction. The mechanism schematized is based on data published by Shiota and colleagues [4]. NUT, which is specifically expressed in spermatids, recruits CBP/p300, which are already expressed in all spermatogenic cell types. NUT stimulates their catalytic activity leading to the acetylation of H4 on both its lysines 5 and 8, which is required for the binding of the first bromodomain of BRDT. The binding of BRDT to H4K5acK8ac leads to the replacement of histones by protamines and the final compaction of the haploid male genome.

**Figure 3 cancers-14-02234-f003:**
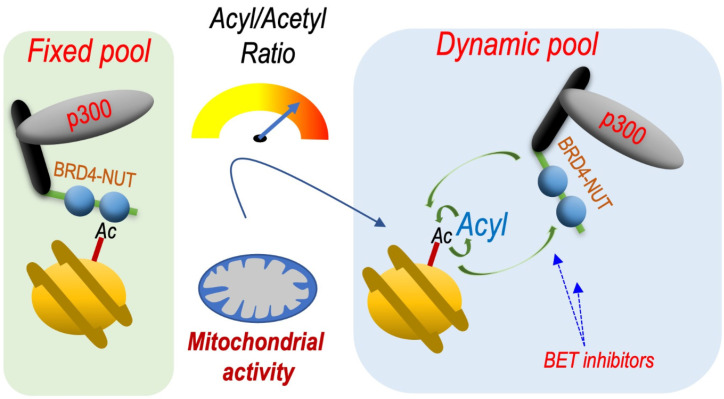
Metabolically controlled variations of the H4K5 acylation/acetylation ratio regulate the dynamics of BRD4-NUT chromatin binding. The mechanism schematized is based on data published by Gao and colleagues [26]. The activation of mitochondrial activity, especially an enhanced fatty acids beta-oxidation, leads to increased levels of butyrylation and crotonylation of H4 lysine 5 compared to its acetylation. Since BRD4 bromodomains are unable to bind H4K5bu or H4K5cr, the relative increase of these two modifications loosens the interaction between BRD4 and acetylated chromatin, leading to an increased dynamics of BRD4-NUT binding in the foci. Gao and colleagues showed that, under these conditions, BRD4 becomes more sensitive to the BET bromodomain small molecule inhibitor, JQ1. Therefore, based on these data, one can speculate that enhancing different metabolic pathways favouring histone >3 carbon chain acylations, would make BRD4-NUT expressing NC tumours more sensitive to a BET inhibitor treatment.

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
