# Peer review of "NUT Is a Driver of p300-Mediated Histone Hyperacetylation: From Spermatogenesis to Cancer"

_cancers, 2022, doi:10.3390/cancers14092234_

Round 1

Reviewer 1 Report

The review manuscript (Manuscript cancers-1682325) entitled “NUT is a driver of p300-mediated histone hyperacetylation: from spermatogenesis to cancer” by Dr. Rousseaux gives a brief overview on the implication of p300-mediated histone acetylation in the context of spermatogenesis. The manuscript looks more like an “opinion paper” rather than a review, while a large number of paragraphs and sections mostly describes authors’ previous findings on and conclusions. For instance “For more than two decades, our laboratory has been following a research program aiming at…”. I suggest increasing the scientific/English sound of the ms by modifying the work in order to reach a more scientific writing style and avoiding these kind of “personal” sentences. However, the ms presents detailed information on the implication of NUT on histone-to-protamine replacement during sperm cell differentiation. Considering the aforementioned aspects, I therefore recommend a major revision. I have several suggestions for improving the manuscript:

General  comments
1.    I suggest increasing the scientific/English sound of the ms by modifying the work in order to reach a more scientific writing style and avoiding “personal” sentences such as ““For more than two decades, our laboratory has been following a research program aiming at…” or “To test our hypothesis, we generated Nut ko mice. As expected, we found..”. A more formal writing style is suggested
2.    All acronyms should be detailed. For instance, among others, BRD3, which is Bromodomain-containing protein 3, HDAC, which is histone deacetilases etc..please revise the entire text accordingly.
3.    General notions on the histone acetilases/deacetilases and their main a.a. targets on histone tails should be included as a background
4.    I suggest including more supporting references as vast parts of sections 3, 4 and 5, especially the first 5-6 sentences of sections 3 and 4, are lacking in references
5.    Figure mentions througout the text should be black instead of blue
6.    Spaces between sentences should be removed

Minor observations
Abstract
1.    “packaging of the male genome” better “sperm cell genome”
2.    “oncogenic BRD4-NUT fusion protein in cancer cells” please detail the cancer type

Introduction
1.    “(please see [6]” I suggest removing please see

Section 2
1.    Spermatogenesis also encompasses extensive modifications in DNA methylation levels, as described in detail here (DOI: 10.3389/fcell.2021.689624). Sentences on the role of DNA methylation changes during sperm cell development and supporting reference should be included. In addition other supporting reference on the sperm cell development, such as PMID: 29525406 and PMID: 33665191
2.    References on histones, nucleosomes and histone-to-protamine transition should be included
Section 3
1.    Is the ref [9], supporting the data on the the KO experiments described in the first paragraph? If yes, the ref should also be mentioned there. Alternatively please included the appropriate reference
Section 5
1.    “(Figure 2).” Should be black

Author Response

The review manuscript (Manuscript cancers-1682325) entitled “NUT is a driver of p300-mediated histone hyperacetylation: from spermatogenesis to cancer” by Dr. Rousseaux gives a brief overview on the implication of p300-mediated histone acetylation in the context of spermatogenesis. The manuscript looks more like an “opinion paper” rather than a review, while a large number of paragraphs and sections mostly describes authors’ previous findings on and conclusions. For instance “For more than two decades, our laboratory has been following a research program aiming at...”. I suggest increasing the scientific/English sound of the ms by modifying the work in order to reach a more scientific writing style and avoiding these kind of “personal” sentences. However, the ms presents detailed information on the implication of NUT on histone-to-protamine replacement during sperm cell differentiation. Considering the aforementioned aspects, I therefore recommend a major revision. I have several suggestions for improving the manuscript:

Reply:

As proposed to and agreed by the guest editor, the focus of this review is the function of NUT in its physiological context of expression, in post-meiotic spermatogenic cells.

By doing so we knew that we will find ourselves in the unusual situation where we had to describe and discuss mostly our own work, since this laboratory is the only one who carried out published and unpublished research on the subject.

Additionally, our will to consider the function of NUT in NC, and later in spermatogenic cells, is directly linked to a very specific research strategy corresponding to the long-term aim of our research program which is deciphering the molecular basis of histone hyperacetylation associated with histone-to-protamine replacement. In particular, this research was motivated by our structural and functional studies of BRDT, a testis-specific BRD4 like protein, as discussed in the text of this review.

This research program and the underlying strategy are very specific to our laboratory. This is the reason why some statements in this review may have appeared as “personal sentences”. In the light of this referee’s remarks, we have made changes throughout the manuscript in an attempt to tone down these “personal sentences” and to use a more neutral tone.

However, we also think that the originality of our research strategy deserves to be highlighted. Indeed, although our research program is aiming at the understanding of the molecular basis histone hyperacetylation in spermatids, we decided to make a “detour” by investigating BRD4-NUT in NUT carcinoma. We think that this approach could be of particular interest to students/post-doc and junior PIs and, consequently, if the editor agrees, we would like to keep some of the narrative style of this manuscript. 

General comments

  1. I suggest increasing the scientific/English sound of the ms by modifying the work in order to reach a more scientific writing style and avoiding “personal” sentences such as ““For more than two decades, our laboratory has been following a research program aiming at...” or “To test our hypothesis, we generated Nut ko mice. As expected, we found..”. A more formal writing style is suggested.

Reply:

As mentioned above, changes were made accordingly throughout the manuscript, including in the cases highlighted by the referee as well as throughout the text.

2- All acronyms should be detailed. For instance, among others, BRD3, which is Bromodomain- containing protein 3, HDAC, which is histone deacetilases etc..please revise the entire text accordingly.

Reply:

We thank the referee for this remark, and have tried to better define the names of the factors mentioned in the text.

3 - General notions on the histone acetilases/deacetilases and their main a.a. targets on histone tails should be included as a background.

Reply:

To the best of our knowledge, there are about 17 deacetylases (4 class I, 6 class II and IV, 7 class III) as well as 17 acetyltransferases (cytoplasmic, 2; GNAT, 2; MYST, 5; p300/CBP, transcriptional coactivators, 2, steroid receptor co-activators, 4) acting on different targets in various contexts. We think that a description of acetylases/deacetylases would therefore take us far beyond the scope of this review. There are several excellent reviews on HATs and HDACs, which discuss all these enzymes and their lysine targets on histones.

4 - I suggest including more supporting references as vast parts of sections 3, 4 and 5, especially the first 5-6 sentences of sections 3 and 4, are lacking in references.

Reply:

We thank the referee for this suggestion and have now added appropriate references in support of all the sentences in the indicated sections

5- Figure mentions througout the text should be black instead of blue.

Reply:  Done

6- Spaces between sentences should be removed.

Reply: Done

Minor observations

Abstract

  1. “packaging of the male genome” better “sperm cell genome”.

Reply: Done

  1. “oncogenic BRD4-NUT fusion protein in cancer cells” please detail the cancer type.

Reply: Done

Introduction
1. “(please see [6]” I suggest removing please see

Reply:

The sentence seems to be incomplete. We do not understand.

Section 2.

1-Spermatogenesis also encompasses extensive modifications in DNA methylation levels, as described in detail here (DOI: 10.3389/fcell.2021.689624). Sentences on the role of DNA methylation changes during sperm cell development and supporting reference should be included. In addition other supporting reference on the sperm cell development, such as PMID: 29525406 and PMID: 33665191.

Reply:

We think that discussing the changes in DNA methylation levels during spermatogenesis would take us outside the scope of this review. Indeed, the focus of our work was to decipher the molecular basis of histone hyperacetylation associated with histone-to-protamine replacement and, to the best of our knowledge, there is no indication of a link between this histone hyperacetylation and changes in DNA methylation. 

Following this referee’s suggestion regarding the epigenetics of spermatogenesis, we have added references to some general reviews.

2 - References on histones, nucleosomes and histone-to-protamine transition should be included

Reply: Done

Section 3.

1-Is the ref [9], supporting the data on the the KO experiments described in the first paragraph? If yes, the ref should also be mentioned there. Alternatively please included the appropriate reference.

Reply:

The appropriate reference has now been added.

Section 5.

1-“(Figure 2).” Should be black

Reply: Done

Reviewer 2 Report

Dr. Khochbin's Lab have made fundamental contributions to our understanding of the relationship between Nutm1 protein, P300 HAT and histone acetylation marks in development and cancer. This is a comprehensive review about the discovery of the functions of the Nutm1 protein in spermatogenesis and its implication in the cancer biology of Nut carcinoma.  I would recommend accept for publication as it is.

One suggest I would make is it is perhaps interesting to elaborate and speculate on the implication of the NUT-P300 axis on other NUT fusion oncogenes such as YAP-NUTM1, CIC-NUTM1 etc., where the positive feedback loop that causes the expansion of the histone acetylation domains might not be in play in the future perspective section. 

Author Response

Dr. Khochbin's Lab have made fundamental contributions to our understanding of the relationship between Nutm1 protein, P300 HAT and histone acetylation marks in development and cancer. This is a comprehensive review about the discovery of the functions of the Nutm1 protein in spermatogenesis and its implication in the cancer biology of Nut carcinoma. I would recommend accept for publication as it is.

One suggest I would make is it is perhaps interesting to elaborate and speculate on the implication of the NUT-P300 axis on other NUT fusion oncogenes such as YAP-NUTM1, CIC- NUTM1 etc., where the positive feedback loop that causes the expansion of the histone acetylation domains might not be in play in the future perspective section.

Reply: We thank the reviewer for the her/his positive evaluation of this review.

Following the reviewer’s suggestion on the implication of the NUT-P300 axis on other NUT fusion oncogenes, we have now added a new paragraph speculating on this point in the “Future perspective” section.
